# Structural elements of a pH-sensitive inhibitor binding site in NMDA receptors

Michael C. Regan[1], Zongjian Zhu[2,3], Hongjie Yuan[2], Scott J. Myers[2], Dave S. Menaldino[4], Yesim A. Tahirovic[4], Dennis C. Liotta[4], Stephen F. Traynelis[2] & Hiro Furukawa[1]

Context-dependent inhibition of N-methyl-D-aspartate (NMDA) receptors has important therapeutic implications for the treatment of neurological diseases that are associated with altered neuronal firing and signaling. This is especially true in stroke, where the proton concentration in the afflicted area can increase by an order of magnitude. A class of allosteric inhibitors, the 93-series, shows greater potency against GluN1-GluN2B NMDA receptors in such low pH environments, allowing targeted therapy only within the ischemic region. Here we map the 93-series compound binding site in the GluN1-GluN2B NMDA receptor amino terminal domain and show that the interaction of the N-alkyl group with a hydrophobic cage of the binding site is critical for pH-dependent inhibition. Mutation of residues in the hydrophobic cage alters pH-dependent potency, and remarkably, can convert inhibitors into potentiators. Our study provides a foundation for the development of highly specific neuroprotective compounds for the treatment of neurological diseases.

[1] WM Keck Structural Biology Laboratory, Cold Spring Harbor Laboratory, Cold Spring Harbor, NY 11724, USA. [2] Department of Pharmacology, Emory University School of Medicine, Atlanta, GA 30322, USA. [3] Department of Neonatology, First Affiliated Hospital of Xi'an Jiaotong University, 710061 Xi'an, Shaanxi, China. [4] Department of Chemistry, Emory University, Atlanta, GA 30322, USA. Correspondence and requests for materials should be addressed to H.F. (email: furukawa@cshl.edu)

In the ischemic environment induced by stroke, the pH of the afflicted area can drop dramatically from 7.4 to as low as 6.5 as a result of altered metabolism and the concurrent buildup of carbon dioxide due to reduced tissue perfusion[1–4]. The extracellular pH falls during ischemia and traumatic brain injury, which also triggers glutamate release by a variety of mechanisms, resulting in hyperactivity of N-methyl-D-aspartate (NMDA) receptors. Overactive NMDA receptor-mediated cation entry into the neuron leads to neuronal death[1,5], a process often referred to as excitotoxicity[6]. The NMDA receptor is a ligand-gated ion channel with key roles in learning and memory and has also been implicated in a number of neurological diseases and disorders, including schizophrenia[7], epilepsy[8], and depression[9]. The majority of NMDA receptors reside in the cell membrane as a heterotetramer consisting of two GluN1 subunits with a and b isoforms determined by an alternatively spliced exon 5, and two GluN2 subunits, of which there are four different subtypes (GluN2A–D)[10]. Three structured domains comprise the intact NMDA receptor: the regulatory amino-terminal domain (ATD), the ligand-binding domain (LBD), and the channel-forming transmembrane domain (TMD)[11–15].

Subtype-selective small-molecule inhibitors of the NMDA receptor are desirable, as they can effectively reduce excitotoxicity during ischemia in preclinical models (see Table S2 in ref.[4]). Previous studies have suggested that GluN2B-containing NMDA receptors can more effectively cause excitotoxic neuronal death compared to GluN2A-containing NMDA receptors[16,17], highlighting the relevance of targeting GluN2B-containing NMDA receptors as a potential therapeutic strategy for ischemic injuries. A wide array of competitive antagonists have been described that bind at the LBD[18–22], but this domain bears high sequence identity among all of the glutamate-binding GluN2 subunits, which could potentially allow for non-specific effects through block of all NMDA receptors[10,23]. The composition of the ATD, by contrast, is highly divergent among the GluN2 subtypes, and is therefore an attractive target for subtype-specific negative allosteric modulation[23,24]; indeed, the phenylethanolamine class of allosteric NMDA receptor inhibitors, which includes the iconic compounds ifenprodil, eliprodil, and the closely related acetamide radiprodil, bind specifically to the GluN1–GluN2B ATD heterodimer interface in GluN2B-containing NMDA receptors[25]. Recent structural work has shown that these small molecules physically cannot be accommodated within the GluN1–GluN2A ATD heterodimeric interface in GluN2A-containing NMDA receptors[26].

The development of allosteric inhibitors that exert a stronger inhibitory effect at low pH have been shown to be neuroprotective during ischemic events[27], and a novel class of GluN2B-specific compounds referred to as the 93-series exhibit highly pH-dependent inhibition of NMDA receptors[4,28], but until now the underlying molecular basis for this effect was unknown. In order to better understand the context-dependence of these compounds as a means to designing effective therapeutics, we employ X-ray crystallography, isothermal titration calorimetry (ITC), and two-electrode voltage clamp (TEVC) electrophysiology to determine where and how this class of inhibitors binds to the NMDA receptor. Here, we show that the pH-dependent inhibitory effect of the 93-series compounds is partly attributed to direct binding to the GluN1b–GluN2B heterodimeric interface and that the placement of the N-alkyl group to the sub-pocket which we name the hydrophobic cage is critical for pH-dependency. Extensive site-directed mutagenesis indicate that the protonation state of a histidine residue in the hydrophobic cage controls the strength of hydrophobic interactions with the N-alkyl group.

## Results

**pH-sensitive binding affinity of the 93-series.** The 93-series compounds are allosteric inhibitors of GluN1–GluN2B NMDA receptors, which bind to the GluN1–GluN2B subunit interface within the ATD. These compounds were designed based on the propanolamine AM-92016 and show up to 10-fold higher potency at pH 6.9 as compared to pH 7.6 (refs.[4,28]). We focused on a subset of the 93-series compounds with systematic structural diversity, ranging from an unsubstituted analog to an N-butyl substitution (Fig. 1a). Yuan et al.[4] recently demonstrated that the N-butyl substitution, 93-31, showed a large pH-induced potency enhancement, which they referred to as the pH-boost and which we confirmed using TEVC recordings in Xenopus laevis oocytes (Fig. 1b). 93-31 at half-maximal inhibitory concentrations did not alter glutamate potency at pH 7.6 or 6.9, eliminating reduced activation by glutamate as a potential component of pH sensitivity (Supplementary Figure 1a). The $IC_{50}$ values for 93-31 inhibition of exon 5-lacking GluN1a–GluN2B receptors shifted from $1.7 \pm 0.38\ \mu M$ at pH 7.6 to $0.23 \pm 0.05\ \mu M$ at pH 6.9—a pH-boost of 7.4 per half log change in extracellular pH (Fig. 1c; Table 2). $IC_{50}$ values were virtually identical for exon 5-containing GluN1b–GluN2B receptors and showed a pH-boost of 9.4 from $1.7 \pm 0.26\ \mu M$ at pH 7.6 to $0.18 \pm 0.05\ \mu M$ at pH 6.9 ($n = 9$), suggesting the inclusion of exon 5 does not influence pH sensitivity of this class of modulators (Supplementary Figure 1b). This observation is consistent with the previous finding that the location of the exon 5-encoded motif is at the domain interface between the ATD and the LBD in the intact NMDA receptors[29], which is too far away from the allosteric inhibitor-binding site to have any physical impact on inhibitor binding[25].

In order to directly estimate binding affinities of these inhibitors for their binding site in the GluN1–GluN2B NMDA receptors, we performed ITC using the purified GluN1a–GluN2B ATD proteins titrated with ligands of various N-substituent lengths at pH 7.6 and pH 6.5 to mimic the extreme drop in pH that can occur at the ischemic core[1–4] (Fig. 2a, b; Supplementary Figures 2 and 3). Toward this end, we expressed the isolated rat GluN1a–GluN2B ATD heterodimer proteins in GnTI⁻ HEK293S cells. Here, for clarity we will call the GluN1 ATD with and without exon 5 GluN1b ATD and GluN1a ATD, respectively. For all of the compounds tested, the ITC experiments unambiguously detected heat exchange when they are injected into the ATD protein samples (Fig. 2). As expected, ifenprodil, which does not have an analogous N-alkyl substitution, did not exhibit a detectable pH-dependent binding affinity, with a $K_D$ of $226 \pm 15\ nM$ at pH 7.6 and a $K_D$ of $249 \pm 16\ nM$ at pH 6.5, corresponding to a negligible pH-boost (Fig. 2c; Table 1). Consistent with the previous report[4], several of the 93-series compounds with relatively small N-alkyl substitutions also showed negligible differences in binding affinities, including the unsubstituted 93-4, N-ethyl 93-5, N-propyl 93-6, and N-isopropyl 93-115 (Figs. 1a and 2c, d). Indeed, not until the volume of the N-alkyl substituent is greater than a propyl substitution (approximately 50 Å³) does the pH-sensitive binding affinity become apparent, rapidly increasing to induce a pH-boost of 1.6 for the N-butyl 93-31 and 2.1 for N-methylpropyl 93-97 (Fig. 2d; Table 1).

**Structure of the ATD heterodimer bound to the 93-series.** As a means to better understand the precise molecular interactions

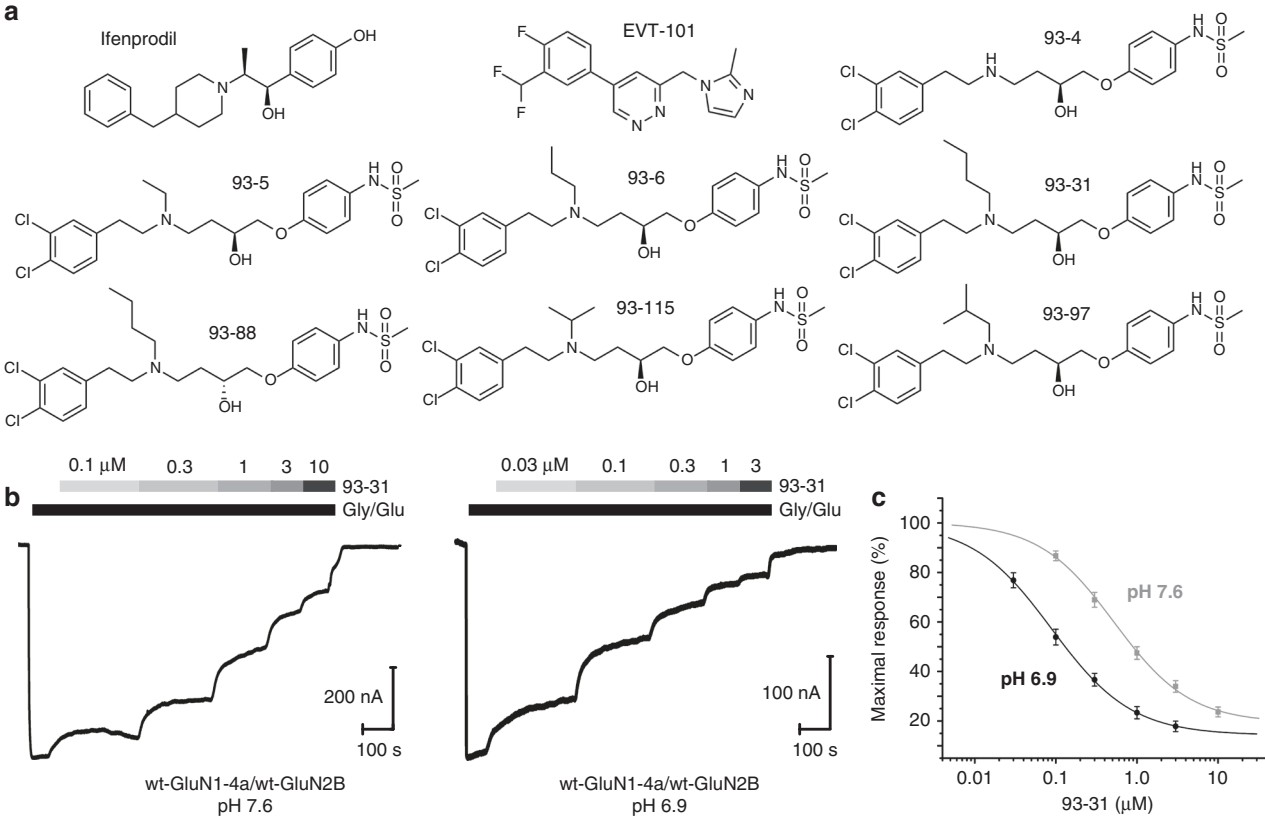

**Fig. 1** pH-sensitive negative allosteric modulators of GluN2B-containing NMDA receptors. **a** The 93-series compounds[28] are shown together with the phenylethanolamine ifenprodil and EVT-101. **b** Representative TEVC recordings of wild-type GluN1-4a/GluN2B NMDA receptors in *Xenopus* oocytes are shown in response to maximally effective concentration of glutamate and glycine (100 and 30 μM, respectively). When normalized to the maximal response, recordings at pH 6.9 showed substantially higher potency of 93-31 than at pH 7.6. **c** Concentration–response curves from TEVC experiments at pH 7.6 (gray) and 6.9 (black) for inhibition of wild-type GluN1-4a/GluN2B NMDA receptor by 93-31 (also see Table 2). Symbols and error bars represent mean ± S.E.M.; the number of replicates is listed in Table 2

taking place between the ATD and the inhibitory ligands, we obtained X-ray crystallographic structures of the 93-series compounds bound to the GluN1–GluN2B ATD heterodimer. Despite exhaustive efforts, we were not able to consistently produce high-quality crystals of the GluN1a–GluN2B ATD complex; we therefore revisited and improved our previous method to use the *Xenopus* GluN1b ATD and rat GluN2B ATD[25], since this splice variants showed identical potency and pH sensitivity as GluN1a. As described in Methods, we were able to streamline and optimize our purification and crystallization conditions in order to reliably produce large crystals of the GluN1b–GluN2B inhibitor complex which routinely diffracted considerably better than in previous studies[25,30], up to 2.1 Å (Supplementary Table 1); ITC experiments confirmed that the two constructs have nearly identical binding properties for ifenprodil (Table 1; Supplementary Figure 4). All of the crystal structures showed unambiguous density for the GluN1b and GluN2B ATD proteins as well as the tested ligands at the inter-subunit interface of the GluN1b–GluN2B ATD heterodimers (Supplementary Figures 5 and 6). The structure of the GluN1b–GluN2B ATD heterodimers is superimposable to that of the GluN1a–GluN2B ATD heterodimers within the GluN1a–GluN2B heterotetrameric NMDA receptor channel as shown previously[11]. Furthermore, the 21 residues encoded by exon 5 in GluN1b are distantly located from the allosteric modulator binding sites. Thus, the structural information of the compound binding site obtained in GluN1b–GluN2B

ATD is equivalent to that in the GluN1a–GluN2B ATD[25], consistent with our functional data showing identical sensitivity of both splice variants to 93-31 at all pH values tested.

The binding site of the 93-series compounds overlays closely with the canonical phenylethanolamine-binding site at the GluN1b–GluN2B subunit interface (Fig. 3a–e). However, the binding mode is quite different, as the backbone of the 93-series ligands adopts a unique "Y-shaped" conformation compared to the more linear arrangement of ifenprodil (Fig. 3f). Furthermore, the binding mode of the NMDA receptor inhibitor EVT-101 (ref. [30]) overlaps with the positioning of the 93-series dichlorophenyl group and the N-alkyl group (Fig. 3g). This series therefore appears to be the first that captures all interactions observed in the three parts of the ifenprodil pocket, in that it overlaps both with ifenprodil and EVT-101. The alkyl-substituted amine of the 93-series compounds forms a hydrogen bond with GluN2B(Gln110), while the dichlorophenyl group is favorably positioned to form hydrophobic contacts with GluN1b(Phe113), GluN2B(Pro177), GluN2B(Ile111), and GluN2B(Phe114) (Fig. 3d, e). The arylsulfonamide group lies at the opposite end of the binding pocket, where it forms hydrogen bonds with GluN2B (Glu236) and with the backbone amides of GluN2B(Met207) and GluN2B(Ser208) (Fig. 3d, e). The N-alkyl substitution of the 93-series compounds branches into the extended binding site and forms van der Waals interactions with GluN1b(Tyr109), GluN1b (Ile133), GluN2B(Met134), and GluN2B(Pro177) (Figs. 3e and 4a). The extent of the van der Waals contacts in this site depends on the orientation and the size of the N-alkyl group of the 93-

series compounds. Among all of the 93-series compounds tested, the N-butyl group of 93-31 most closely matches the shape of the hydrophobic cage by aligning in such a way as to form a

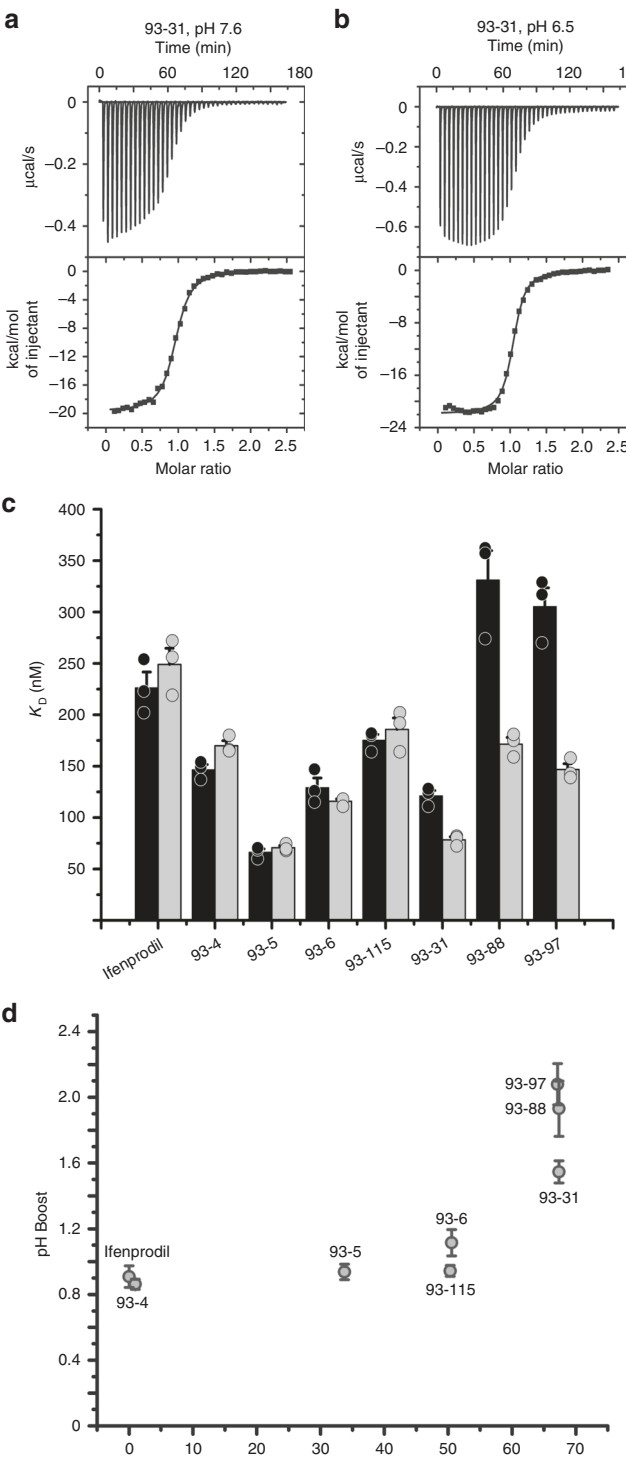

**Fig. 2** ITC titrations of the GluN1/GluN2B ATD with different ligands at pH 7.6 and 6.5. Representative ITC isotherms of the isolated GluN1-GluN2B ATD titrated with 93-31 at pH 7.6 (**a**) and 6.5 (**b**). The average dissociation constant ($K_D$) for 93-31 at pH 7.6 was 121 ± 5.1 nM, which drops to 78.4 ± 2.97 nM at pH 6.5 (Table 1). **c** Dissociation constants for various 93-series ligands at pH 7.6 (black bar) and pH 6.5 (gray bar). **d** The ratio of the $K_D$ at pH 7.6 to the $K_D$ at 6.5 (pH-boost) is shown as a function of N-substituent volume. For all panels, error bars represent mean ± S.E.M. All experiments were performed in triplicate

hydrophobic contact with the side chain of GluN1b(Ile133) (Supplementary Figure 7).

These binding modes are consistent with many of the trends that appear in our ITC data. The unsubstituted variant 93-4 has a weaker $K_D$ at high and low pH than the N-methyl 93-5, N-propyl 93-6, and N-butyl 93-31 variants because it lacks the aliphatic substituent required to interact with the protein residues in the extended binding site. As the N-alkyl substituent grows to the size of N-butyl, it is finally long enough to form these additional interactions to allow for tighter binding at the extended site. This is reflected in higher affinities than with the unsubstituted 93-4, as well as ifenprodil (Table 1). The branched N-methylethyl 93-115 and N-methylpropyl 93-97 variants, which do not have a larger volume than the N-butyl compound, likely suffer from steric constraints not encountered by the higher-affinity unbranched 93-31. All three compounds tested which had sufficient N-alkyl volumes to fill the extended binding site (93-31, 93-88, and 93-97) showed a considerable pH-boost when measured by ITC. N-methylpropyl (93-97) shows a large boost with respect to pH but weaker absolute binding affinity than 93-31, again, possibly due to steric obstruction. The compound 93-88 is N-butyl substituted like 93-31 and also shows a pH-boost; however, as the (R) enantiomer, 93-88 must adopt quite a different binding pose in order to form the hydrogen bond to GluN2B(Gln110), which in turn contorts the molecule into an unfavorable orientation and reduces binding affinity (Supplementary Figures 6 and 7). These data allow us to interrogate one component—ligand binding to the isolated ATD heterodimer—of the complex pH-dependence of these inhibitors, which may also include pH-dependent changes in the conformations of the full-length protein.

**Functional characterization of the 93-series binding pocket.** To validate the unique binding mode of the 93-series compounds observed in our crystallographic studies and to gain mechanistic insights into the pH sensitivity of the 93-31, we conducted site-directed mutagenesis of residues around the binding pocket and measured the ion channel activity by TEVC recordings in *Xenopus* oocytes. Toward this end, we obtained concentration–response curves of 93-31 at pH 6.9 and pH 7.6 on the mutant GluN1-4a-GluN2B NMDA receptors. We first examined a potential role for the arylsulfonamide group of the 93-series compounds in facilitating pH sensitivity, which interacts with GluN2B(Glu236) (Fig. 3d, e). Although the pKa of similar arylsulfonamide groups[31,32] are often between 6 and 7, analytical evaluation determined the pKa to be 8.0, suggesting only a modest change of the concentration of protonated sulfonamide from 72% at pH 7.6 to 93% at pH 6.9, a shift of 1.3-fold. Any potential change in the ionization of the sulfonamide nitrogen itself as the pH drops from 7.6 to 6.9 might change the nature of the interaction between the ligand and GluN2B(Glu236). However, consistent with the minimal change in ionization of the sulfonamide given the measured pKa value, changing the charge of the Glu236 side chain in GluN2B(Glu236Gln) produced no substantial change in $IC_{50}$ or efficacy of 93-31 inhibition at either pH 6.9 or pH 7.6 as compared to the wild type (Fig. 4c; Table 2). These data confirm that the GluN2B(Glu236)-sulfonamide interaction does not contribute to the pH-sensitive effect.

As pH sensitivity appears to be a function of 93-series N-substituent volume, we next tested residues in the immediate vicinity of the N-alkyl group within the protein hydrophobic cage. As described above, our crystal structures clearly identified GluN1b(Ile133) as the major interacting residue with the N-alkyl group of the 93-series compounds, and the strength of van der Waals interactions with GluN1b(Ile133) will vary depending on the nature of the N-alkyl substituent (Fig. 4a, b; Supplementary

**Table 1 Results of ITC experiments titrating the ATD dimer with various ligands**

| Ligand | Ifenprodil (GluN1a–GluN2B) | Ifenprodil (GluN1b–GluN2B) | 93-4 | 93-5 | 93-6 | 93-115 | 93-31 | 93-88 | 93-97 |
|---|---|---|---|---|---|---|---|---|---|
| N-substituent volume (Å³) | (n/a) | (n/a) | 0.97 | 33.74 | 50.54 | 50.33 | 67.35 | 67.35 | 67.13 |
| $n^{pH\ 7.6}$ | 1.0 ± 0.01 | 0.98 ± 0.01 | 0.98 ± 0.02 | 1.00 ± 0.01 | 1.0 ± 0.01 | 1.0 ± 0.01 | 0.95 ± 0.01 | 0.99 ± 0.02 | 1.0 ± 0.02 |
| $n^{pH\ 6.5}$ | 0.97 ± 0.01 | – | 0.98 ± 0.01 | 0.98 ± 0.01 | 1.0 ± 0.01 | 1.0 ± 0.01 | 1.0 ± 0.01 | 1.00 ± 0.02 | 0.99 ± 0.02 |
| $K_D^{pH\ 7.6}$ (nM) | 226 ± 15 | 255 ± 11 | 147 ± 5.0 | 66.3 ± 3.2 | 129 ± 9.3 | 175 ± 5.6 | 121 ± 5.1 | 331 ± 29 | 305 ± 18 |
| $K_D^{pH\ 6.5}$ (nM) | 249 ± 16 | – | 170 ± 4.8 | 70.7 ± 2.0 | 116 ± 2.2 | 186 ± 11 | 78 ± 3.0 | 171 ± 6.6 | 147 ± 5.6 |
| $\Delta H^{pH\ 7.6}$ (kcal mol⁻¹) | −1.53 × 10⁴ ± 346 | −1.48 × 10⁴ ± 265 | −2.37 × 10⁴ ± 457 | −2.28 × 10⁴ ± 834 | −1.94 × 10⁴ ± 809 | −2.38 × 10⁴ ± 708 | −2.16 × 10⁴ ± 944 | −2.79 × 10⁴ ± 765 | −2.12 × 10⁴ ± 568 |
| $\Delta H^{pH\ 6.5}$ (kcal mol⁻¹) | −1.70 × 10⁴ ± 982 | – | −2.24 × 10⁴ ± 483 | −2.11 × 10⁴ ± 1140 | −1.96 × 10⁴ ± 146 | −2.09 × 10⁴ ± 568 | −2.22 × 10⁴ ± 807 | −2.03 × 10⁴ ± 941 | −1.75 × 10⁴ ± 941 |
| $T\Delta S^{pH\ 7.6}$ (kcal mol⁻¹) | −6.10 × 10³ ± 308 | −5.64 × 10⁴ ± 264 | −1.42 × 10³ ± 447 | −1.28 × 10⁴ ± 866 | −9.87 × 10³ ± 804 | −1.41 × 10⁴ ± 536 | −1.19 × 10⁴ ± 914 | −1.89 × 10⁴ ± 804 | −1.22 × 10⁴ ± 544 |
| $T\Delta S^{pH\ 6.5}$ (kcal mol⁻¹) | −7.84 × 10³ ± 960 | – | −1.30 × 10³ ± 500 | −1.12 × 10⁴ ± 1140 | −9.92 × 10³ ± 132 | −1.16 × 10⁴ ± 604 | −1.23 × 10⁴ ± 811 | −1.09 × 10⁴ ± 1440 | −8.00 × 10³ ± 919 |
| pH-boost | 0.91 ± 0.066 | – | 0.86 ± 0.030 | 0.94 ± 0.047 | 1.11 ± 0.081 | 0.94 ± 0.034 | 1.55 ± 0.067 | 1.93 ± 0.168 | 2.08 ± 0.125 |

Respective ligands were titrated into the GluN1a–GluN2B ATD heterodimer. Superscript indicates the pH at which the experiments were performed to generate the given value. pH-boost is the ratio of $K_D^{pH\ 7.6}$ to $K_D^{pH\ 6.5}$. The isolated GluN1a-GluN2B ATD expressed in HEK cells was used for all ITC experiments, except for one set of experiments in which we titrated ifenprodil into the isolated GluN1b–GluN2B ATD used for crystallization. Values are presented as mean ± S.E.M. All experiments were performed in triplicate as described in Methods. Side chain volumes were calculated using MolInspiration (molinspiration.com). n/a not applicable

Figure 7). The GluN1b(Ile133Ala) mutation, which decreases the volume of the hydrophobic side chain, consistently reduces both potency and efficacy of inhibition by 93-31 as well as the pH-boost (Fig. 4d; Table 2). The placement of the N-alkyl substituent is also likely to be stabilized by the side chain of GluN2B(Met134) and GluN2B(Pro177), which are located within 5 Å of the N-alkyl group. The GluN2B(Met134Ala) mutation does not affect potency, but increases the current at pH 6.9 (36%) and reduces pH-boost (3.2-fold). The GluN2B(Pro177Gly) mutation had a robust effect on potency, efficacy, and pH-boost (Fig. 4e, f; Table 2).

Finally, the hydrophilic residue sitting at the edge of the pocket, GluN2B(Asp136) (Fig. 4a, b), is not directly interacting with the N-alkyl group. Reduction of hydrophilicity by the GluN2B (Asp136Ala) mutation had a minor increase in pH-boost with minimal changes in potency and efficacy, perhaps reflecting slightly favored positioning of the N-alkyl group in the pocket (Fig. 4g; Table 2). Overall, the above mutagenesis studies indicate that all of the residues contributing to hydrophobic interactions with the N-alkyl group of the 93-31 are involved in controlling efficacy, potency, and pH-boost.

Because the chemistry of pH sensitivity typically involves changes in protonation states associated with altered solution pH, we speculated that the hydrophobic interaction involving the N-alkyl group of the 93-series compounds is controlled by the protonation of residues near the hydrophobic pocket. One such residue that caught our attention was GluN1b(His134), which is positioned to form a hydrophobic interaction with GluN1b (Ile133) when it is not protonated (Fig. 4b). When protonated at low pH, GluN1b(His134) has a considerably weaker hydrophobic interaction with GluN1b(Ile133), which in turn allows GluN1b (Ile133) to form stronger interactions with the N-alkyl group of the 93-series compounds and serve as a local pH sensor. GluN1b (His134) is located at the exit of the GluN1b–GluN2B subunit interface and is solvent accessible, allowing it to sense solution pH. The pKa of His is 6 in free solution, and likely different in the context of adjacent residues. Thus, the His134 pKa could be tuned to show a maximal change in the ionization state with changes in extracellular pH. For example, a pKa of 6.4 would yield a four-fold change and a pKa of 6.8 would yield a three-fold change in the ionization state of His134 with a change in extracellular pH from 7.6 to 6.9. Further supporting the important role of the protonation state of GluN1(His134) is the observation that breakage of the GluN1(His134)–GluN1(Ser108) interaction by

mutating GluN1(Ser108Ala) robustly reduces 93-31 potency, efficacy, and pH-boost (Fig. 4i; Table 2). Situated in the middle of these pH-sensitive elements is GluN1(Tyr109), proximal to the backbone and the N-alkyl group of the 93-series compounds. The GluN1(Tyr109Ala) mutation to remove the bulky side chain mildly lowers potency and efficacy of inhibition, but not pH-boost (Table 2).

We tested the GluN1a(His134Ala) mutation[4] prior to our current crystallographic studies and not only observed elimination of pH-boost (as predicted with the loss of ionizeable side chain), but also the conversion of 93-31 from an inhibitor to a potentiator (Fig. 4h; Table 3). A similar potentiating effect of the GluN1a(His134Ala) mutation was observed for 93-5, whereas we observed only a minor effect for ifenprodil (Supplementary Figure 8a, b), indicating that the potentiating effect is facilitated by the N-alkyl group of the 93-series compounds. GluN1b (His134) is thus a critical regulator of pH-sensitive compound binding and allosteric inhibition in general. Analogous to mutations at His134, we observed that substituting a bulkier group for GluN1(Tyr109Trp) likely disrupted the local architecture of the pH-sensitive elements, and had a similar effect to the GluN1 (His134Ala) mutation, in that binding of 93-31 and 93-5 results in potentiation rather than inhibition. As in the case of GluN1 (His134Ala), only minor potentiation was observed for ifenprodil, confirming the role of the 93-series N-alkyl group in this effect. Neither of these potentiating mutations altered the receptor's sensitivity to extracellular protons (Supplementary Figure 8e, f; Table 4), indicating that the potentiation does not involve a change in tonic proton inhibition. This indicates that the architecture of the binding pocket for the N-alkyl group and its vicinity is uniquely organized to mediate pH-sensitive compound binding. Furthermore, the area around GluN1 (His134) is a local pH sensor that can potentially be manipulated in order to develop pH-dependent positive and negative allosteric modulators (Fig. 4h; Supplementary Figure 8c; Table 3).

## Discussion
Our work here has identified a ligand-binding mode specific to GluN2B-containing NMDA receptors that encompasses and expands the canonical phenylethanolamine-binding pocket[25,30]. The 93-series compounds occupy the ifenprodil pocket and also uniquely fill a hydrophobic pocket that branches out from the phenylethanolamine-binding site, thereby introducing a greater

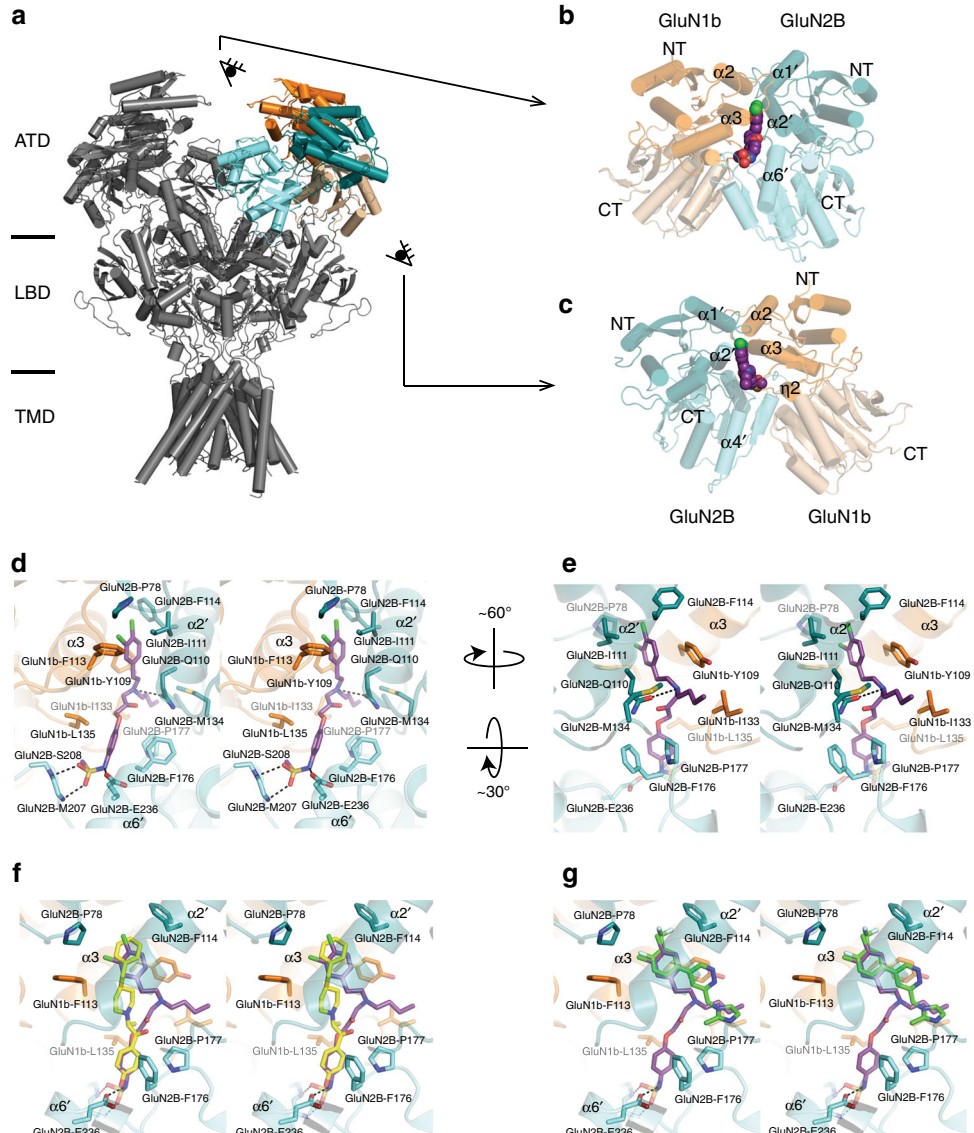

**Fig. 3** Structure of the 93-series binding site. **a** The intact heterotetrameric GluN1b/GluN2B NMDA receptor[34] is composed of three structured domains, with the ATD furthest from the cell membrane (PDB code: 6CNA). **b**, **c** The crystal structure of the isolated GluN1b/GluN2B ATD heterodimer bound to the pH-sensitive 93-31 reveals the ligand-binding site at the heterodimer interface, as viewed from two angles. **d**, **e** The key residues surrounding the N-alkyl chain of 93-31 are primarily hydrophobic, shown here in stereo view. **f** Overlay of 93-31 and ifenprodil[25] (yellow, PDB code: 3QEL). **g** Overlay of 93-31 and EVT-101 (ref. [29]) (green, PDB code: 5EWM)

number of protein–ligand contacts and thus more options for rational drug design. In this manner, 93-31 is the first ligand to occupy all three elements of the negative allosteric modulatory site within the GluN1–GluN2B ATD heterodimer. Our data also suggest that the 93-series compounds that possess an N-alkyl group that matches the size and shape of the hydrophobic pocket exert allosteric inhibition with substantial pH-dependence. Our ITC experiments demonstrated that the isolated GluN1–GluN2B ATD heterodimer alone shows a pH-boost, although the magnitude of the effect is modest compared to that observed for full-length receptors in TEVC recordings. This difference in the extent of pH-boost may be attributed to the fact that our ITC experiments were conducted on the isolated GluN1–GluN2B ATD heterodimer, whereas TEVC was performed using intact channels that contain two GluN1–GluN2B ATD heterodimers per heteroteramic assembly. We interpret the difference as suggesting that some portion of the pH-boost occurs at the level of compound binding to the ATD but a substantial portion of the

effect occurs through downstream effects perhaps involving conformational alteration in both the LBD and ATD regions[14,33]. Taken together, our experiments demonstrate that the hydrophobic interactions involving the 93-series N-alkyl group are critical to mediating allosteric inhibition in a pH-dependent manner. We predict that the potency and efficacy of inhibition, as well as pH-boost, of the 93-series compounds could be improved by replacing the N-alkyl group with a more extended hydrophobic group, for example, N-aryl, to potentially strengthen hydrophobic interactions with GluN1 (Ile133), GluN2B(Pro177), and GluN2B(Met207).

The structural basis for the pH-sensitive binding of the 93-series compounds may involve GluN1(His134), which is situated next to GluN1(Ile133) and forms a hydrophobic interaction when not protonated at high pH. At lower pH values (higher proton concentrations), however, the interaction between GluN1(Ile133) and GluN1(His134) likely weakens as an increased proportion of GluN1(His134) is protonated. This would be predicted to

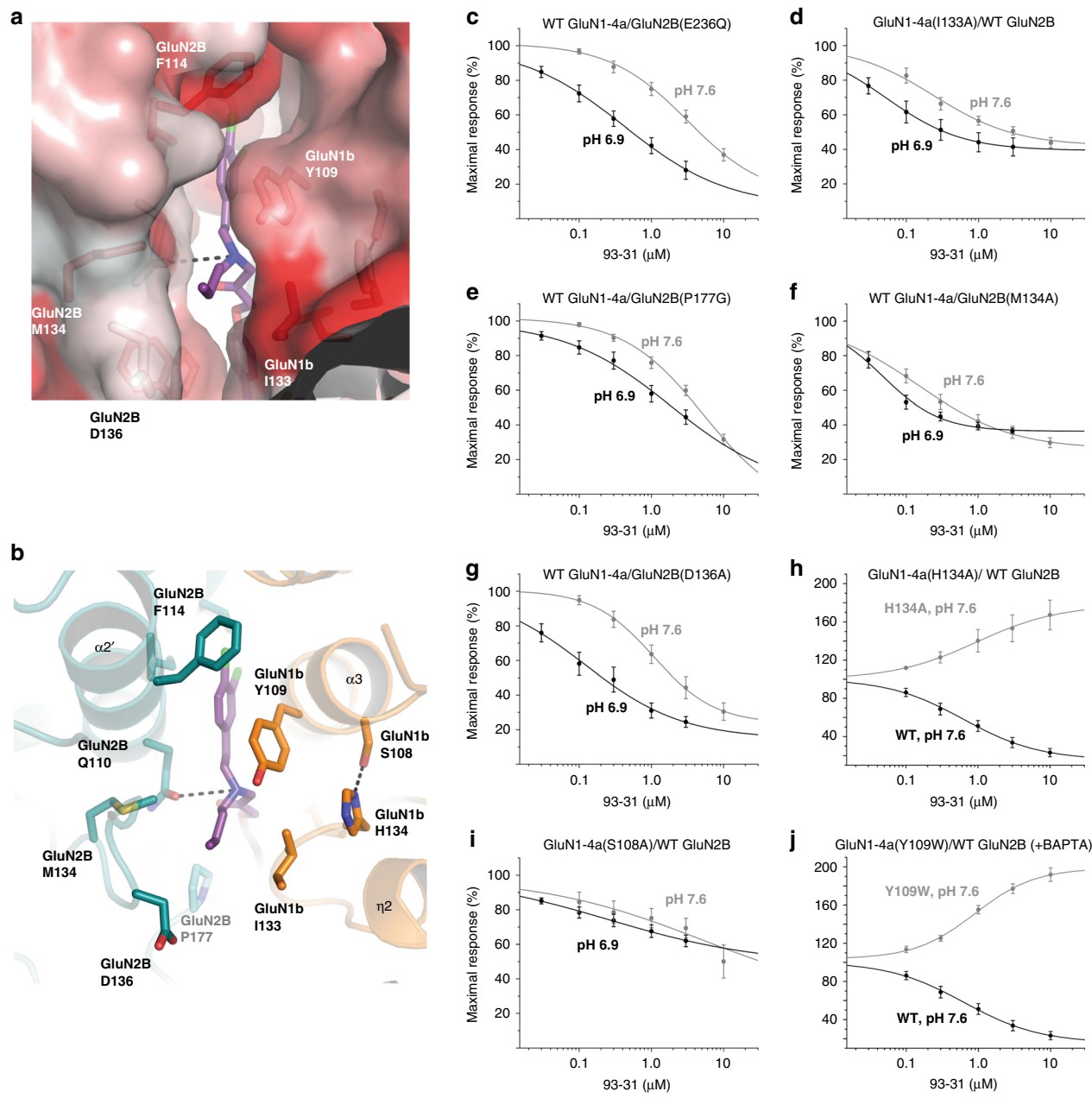

**Fig. 4** TEVC concentration–response curves for GluN1-4a–GluN2B mutants. **a** The N-alkyl group of 93-31 is accommodated by the "hydrophobic cage" as shown by the hydrophobic (red) to non-hydrophobic (white) surface presentation. **b** Key residues at the GluN1/GluN2B ATD dimer interface that interact with the N-alkyl chain of 93-31 were mutated and evaluated for effects on pH sensitivity of 93-31. **c**–**i** Current responses to maximally effective concentration of glutamate and glycine (100 μM glutamate, 30 μM glycine) are shown in the presence of varying concentrations of 93-31 as a proportion of maximal response. Experiments conducted at pH 7.6 are shown in gray; experiments at pH 6.9 are shown in black. **h**, **j** Two mutations, GluN1-4a (His134Ala) and GluN1-4a(Tyr109Trp), convert 93-31 into a potentiator. Experiments here were performed at pH 7.6 (gray), with the wild-type curve displayed for comparison (black). Error bars represent mean ± S.E.M. See Tables 2 and 3 for fitted $IC_{50}$ values and the number of independent replicates

strengthen the van der Waals contacts between the N-alkyl group of the 93-series compounds and GluN1(Ile133).

The GluN1 (His134Ala) mutation[4] not only abolishes the pH-boost but also converts 93-31 and 93-5 into strong potentiators, while only blocking the inhibitory action of ifenprodil. We observed an even greater effect when GluN1(Tyr109)—located next to GluN1(Ile133) and GluN1(His134)—was mutated to a larger tryptophan in order to perturb the positioning of those residues, implying that GluN1(Tyr109), GluN1(Ile133), and GluN1(His134) constitute a local hub that can influence both the binding and downstream action of negative allosteric modulators that bind to the ATD heterodimer interface in GluN1–GluN2B

NMDA receptors. The opening and closing of the bi-lobe structure of GluN2B ATD and rearrangement of the GluN1–GluN2B subunit interface at the ATD is coupled to activation and allosteric inhibition of the GluN1–GluN2B NMDA receptors[14]. The above residues in the hydrophobic cage are located in proximity to the subunit interface, thus are in the ideal position to control the activity of the GluN1–GluN2B NMDA receptors. The potentiating effect of 93-31 and 93-5 on the GluN1(Tyr109Trp) and GluN1(His134Ala) mutants perhaps involves rearrangement of the ATD heterodimers in a way to favor activation, which lead to the tantalizing possibility that GluN2B-specific positive allosteric modulators could be developed. Such modulators could have

therapeutic potential in diseases characterized by NMDA receptor hypofunction, such as schizophrenia[34,35]. Overall, our study shows that by modifying a specific site on this class of ligands within the parameters we have established here, there is great potential for the development of highly targeted therapeutics for neurological diseases.

## Methods

**Materials**. All 93-series compounds were synthesized as described elsewhere[28], and purified to >95% purity. 93-series compounds in this study correspond to the following compounds in Tahirovic et al.[28]: 93-4 is compound 29, 93-5 is compound

67, 93-6 is compound 69, 93-31 is compound 70, 93-88 is compound 106, 93-97 is compound 72, and 93-115 is compound 74. All starting materials and other chemicals were from Sigma unless otherwise indicated. Ifenprodil hemitartrate was purchased from Tocris (#0545).

**Determination of pKa**. The p$K$a values for 93-31 were determined using the spectrometric (UV-metric) technique (Pion Inc., East Sussex, UK, RH18 5DW). 93-31 was titrated in three UV-metric single titrations on SiriusT310026 from pH 12.5–1.5 at concentrations of 62–51 μM, under methanol–water co-solvent conditions (the methanol mixing ratio varied from 60.8 to 34.7% w/w). No precipitation of the sample from solution was observed and two p$K$a values, with aqueous values of 8.0 ± 0.03 and 9.5 ± 0.01, were determined from the spectroscopic data collected by Yasuda–Shedlovsky extrapolation of the individual results obtained[36–38].

**Crystallization of the ATD heterodimer**. Coexpression and purification of the *Xenopus* GluN1b and rat GluN2B ATD heterodimer were performed as described previously[25]. Briefly, *Trichoplusia ni* (High Five, Thermo Fisher) insect cells were infected with a baculovirus harboring both *Xenopus* GluN1b ATD and rat GluN2B ATD for 48 h. The concentrated medium was subjected to purification by metal affinity resin (GE Lifesciences) charged with $CoCl_2$. Poly-Histidine tags at the C-terminus of GluN1b ATD and the N-terminus of the GluN2B ATD were removed by thrombin digestion and the digested samples were further purified by Superdex200 (GE Lifescience). Purified protein was concentrated to 10 mg mL$^{-1}$ and dialyzed against 50 mM NaCl, 10 mM Tris (pH 8.0), and 10 μM ifenprodil hemitartrate (Tocris). Crucially, we found it to be absolutely necessary to filter the dialyzed protein through a 0.1 μm spin filter (Millipore) just before setting up crystal screens. Crystals grew in sodium formate/HEPES as in our previous study[25], initially appearing after 3–4 days, then continuing to grow for up to 2–3 weeks at 18 °C. Crystals were transferred to 2 μL drops containing 4 M sodium formate, 0.1 M HEPES (pH 7.5), 35 mM NaCl, 7 mM Tris (pH 8.0), and 50 μM of the respective inhibitors, and allowed to soak overnight. Crystals were then transferred to a new drop of the same condition and soaked overnight again. In order to gradually increase the concentration of sodium formate for use as a cryoprotectant, these crystals were further transferred to a fresh drop of the same condition but with 4.5 M sodium formate, soaked overnight, and then transferred again to the same condition with 5.0 M sodium formate for a final overnight soak. In combination with the critical protein filtration step, this multi-day gradual soaking protocol was essential to improving crystal diffraction. Crystals were flash-frozen in liquid nitrogen for X-ray diffraction data collection.

**Preparation of the GluN1a–GluN2B ATD fusion proteins**. To efficiently express the ATD domains of *Xenopus* GluN1a and rat GluN2B, we used the alkaline phosphatase secretion signal and fused the two domains with a 57 residue linker as described elsewhere[26], with the following modifications: due to an intrinsic thrombin site in the GluN2B ATD, we engineering Tobacco Etch Virus (TEV) protease sites into this construct to remove the linker. In addition, three glycosylation sites were removed by mutating GluN1a Asn61 and Asn371 to Gln, and GluN2B Asn348 was mutated to Asp.

Suspension cell cultures of HEK 293 S GnTI$^-$ cells (ATCC #3022) were grown in JMEM (VWR), which was modified to include 20% FreeStyle 293 expression media (Gibco), 2.5% FBS, 2.5 mM GlutaMAX (Gibco), 0.3% Primatone, 0.1% Pluronic F68, 20 mM HEPES pH 7.5, and 2 g/L sodium bicarbonate. Cell cultures were maintained at 37 °C with 5% $CO_2$ and grown to a density of

---

### Table 2 Results of TEVC 93-31 concentration–response experiments with GluN1-4a/GluN2B mutants

| Constructs | pH 7.6 $IC_{50}$ (μM), % response[a] $nH$ ($n$) | pH 6.9 $IC_{50}$ (μM), % response[b] $nH$ ($n$) | $IC_{50}$ (pH 7.6)/ $IC_{50}$ (pH 6.9) |
|---|---|---|---|
| GluN1-4a/GluN2B (WT) | 1.7 ± 0.38, 34% $nH$ 0.7 (24) | 0.23 ± 0.05, 18% $nH$ 0.7 (23) | 7.4 |
| GluN1-4b/GluN2B (WT) | 1.7 ± 0.26, 46% $nH$ 1.3 (9) | 0.18 ± 0.05, 22% $nH$ 1.0 (9) | 9.4 |
| GluN1-4a(S108A) | 30 ± 12, 69% $nH$ ND (7) | 20 ± 4.7, 62% $nH$ ND (5) | 1.5 |
| GluN1-4a(Y109A) | 6.2 ± 3.0, 45% $nH$ 0.6 (6) | 0.80 ± 0.30, 28% $nH$ 0.6 (5) | 7.6 |
| GluN1-4a(Y109W) | 1.4 ± 0.37, 186%[c] $nH$ 1.0 (7) | 0.94 ± 0.19, 212%[c] $nH$ 0.8 (8) | 1.5 |
| GluN1-4a(I133A) | 6.3 ± 2.7, 51% $nH$ ND (6) | 1.2 ± 0.42, 41% $nH$ 0.4 (7) | 5.3 |
| GluN2B(M134A) | 1.1 ± 0.44, 36% $nH$ 0.4 (8) | 0.38 ± 0.08, 36% $nH$ 0.4 (8) | 2.9 |
| GluN2B(D136A) | 3.8 ± 1.5, 44% $nH$ 0.8 (6) | 0.36 ± 0.09, 24% $nH$ 0.6 (6) | 11 |
| GluN2B(P177A) | 38 ± 9.7, 73% $nH$ ND (6) | 5.7 ± 1.2, 56% $nH$ ND (4) | 6.7 |
| GluN2B(P177G) | 4.7 ± 0.54, 60% $nH$ ND (9) | 2.3 ± 0.57, 45% $nH$ 0.7 (7) | 2.0 |
| GluN2B(E236A) | 3.2 ± 1.2, 41% $nH$ 0.7 (10) | 0.49 ± 0.10, 22% $nH$ 0.7 (8) | 6.5 |
| GluN2B(E236Q) | 5.2 ± 0.73, 59% $nH$ ND (8) | 0.73 ± 0.17, 28% $nH$ 0.6 (6) | 7.1 |

Concentration–response curves were generated in the presence of 100 μM glutamate and 30 μM glycine, and the listed ligands, and normalized against current from glutamate and glycine alone. $IC_{50}$ values are given ± S.E.M. ($n$): number of independent replicates. In all of the recordings, Hill coefficient ($nH$) ranged from 0.4 to1.3, and were not determined when the inhibition was less than 50% at the highest concentrations tested. ND not determined
[a]The steady-state response was determined as a percentage of the current at 3 μM 93-31, pH 7.6
[b]The steady-state response was determined as a percentage of the current at 3 μM 93-31, pH 6.9
[c]The GluN1-4a (Y109W) mutant converted 93-series ligands into potentiators; $EC_{50}$ value and maximal potentiation given

---

### Table 3 Results of TEVC concentration–response experiments with GluN1-4a/GluN2B potentiating mutants

| Constructs | 93-31 pH 7.6 $IC_{50}$ (μM), % response $nH$ ($n$) | 93-5 pH 7.6 $IC_{50}$ (μM), % response $nH$ ($n$) | Ifenprodil, pH 7.6 $IC_{50}$ (μM), % response $nH$ ($n$) |
|---|---|---|---|
| GluN1-4a/GluN2B (WT)[b] | 1.2 ± 0.46, 34%[a] $nH$ 0.8 (6) | 0.12 ± 0.01, 39%[c] $nH$ 0.6 (5) | 0.12 ± 0.02, 35%[d] $nH$ 1.0 (7) |
| GluN1-4a(Y109W)[b] | 1.3 ± 0.28, 177%[e] $nH$ 1.0 (8) | 0.52 ± 0.20, 265%[e] $nH$ 0.9 (7) | ND, 115%[f] $nH$ ND (9) |
| GluN1-4a(H134A)[b] | 1.8 ± 0.23, 153%[e] $nH$ 1.1 (6) | 0.32 ± 0.08, 140%[e] $nH$ 0.7 (6) | ND, 105%[f] $nH$ ND (9) |

Concentration–response curves were generated in the presence of 100 μM glutamate and 30 μM glycine, and the listed ligands, and normalized against current from glutamate and glycine alone. $IC_{50}$ values are given ± S.E.M. ($n$): number of independent replicates. In all of the recordings, Hill coefficient ($nH$) ranged from 0.6 to 1.1, and were not determined when the potentiation was less than 30% at the highest concentrations tested. ND not determined
[a]The steady-state response was determined as a percentage of the current at 3 μM 93-31, pH 7.6
[b]To prevent run-down of current, experiments in the second table (WT, Y109W, and H134A) were conducted in oocytes co-injected with 13 nL of 100 mM BAPTA
[c]Current at 1 μM 93-5
[d]Current at 1 μM ifenprodil
[e]The GluN1-4a(Y109W) and GluN1-4a(H134A) mutants converted 93-series ligands into potentiators; $EC_{50}$ value and maximal potentiation given
[f]Current at maximal ifenprodil potentiation

**Table 4 Results of TEVC proton concentration–response experiments with inhibiting GluN1-4a/GluN2B mutants**

| Constructs | Proton IC$_{50}$ (nM) ($n$) | pH IC$_{50}$ ($n$) |
|---|---|---|
| GluN1-4a/GluN2B (WT) | 40 ± 1.7 (6) | 7.50 ± 0.02 (6) |
| GluN1-4a(Y109W) | 35 ± 1.7 (8) | 7.55 ± 0.02 (8) |
| GluN1-4a(H134A) | 36 ± 1.4 (7) | 7.54 ± 0.02 (7) |

Concentration–response curves of proton were generated in the presence of 100 µM glutamate and 30 µM glycine. IC$_{50}$ values are given ± S.E.M. ($n$): number of independent replicates. Proton concentration was calculated using a proton activity coefficient of 0.8

$2.0–2.5 \times 10^6$ cells mL$^{-1}$, then infected with baculovirus. Sodium butyrate was added to the infected cells to a final concentration of 5 mM and culture medium containing secreted GluN1a–GluN2B ATD fusion protein was harvested 96 h after infection.

Cells were removed from the medium by centrifugation and the clarified medium was concentrated and exchanged three times to cold TBS (200 mM NaCl, 20 mM Tris, pH 8.0) using tangential flow filtration with a molecular weight cutoff of 30 kDa (Pall Corporation). The fusion protein was purified using metal affinity resin (GE Lifesciences) charged with CoCl$_2$, and the eluted protein was dialyzed to TBS with 40 mM imidazole overnight. The following day, glutathione in a 10:1 ratio, reduced:oxidized, was added to the dialyzed protein to a final concentration of 3 mM, along with EDTA, pH 8.0, to a final concentration of 1 mM. Based on the protein yield, a 1:100 ratio of TEV protease (mass:mass, TEV:ATD) was added to the ATD solution and the reaction was incubated for 4 h at 18 °C. Next, EndoF1 endoglycosidase at a 1:5 ratio (mass:mass, EndoF1:ATD) was added to the ATD/TEV mixture and dialyzed overnight at room temperature to 2 L TBS with 10% glycerol and 20 mM imidazole.

The TEV digested, deglycosylated, dialyzed protein was subjected to a second pass through a cobalt metal affinity column to remove TEV, EndoF1, and the fusion protein linker, and the runoff and washes using TBS with 20 mM imidazole were collected until the OD$_{280}$ reached zero. This sample was then dialyzed overnight at 4 °C to 2 L TBS with 20% glycerol. Finally, the purified ATD heterodimer was concentrated to 5 mg mL$^{-1}$ (45 µM) and flash-frozen in liquid nitrogen, and then stored at −80 °C until needed.

**Determination of ligand-binding affinity by ITC**. Purified GluN1a–GluN2B ATD proteins from the above-described fusion protein purification was dialyzed extensively against a buffer containing 150 mM NaCl, 20 mM sodium phosphate, and 5% glycerol. The final pH of the solution was achieved using a ratio of dibasic: monobasic sodium phosphate for pH values 6.5 and 7.6.

After dialysis, the protein sample was diluted to 10–15 µM and dimethyl sulfoxide (DMSO) was added to a final concentration of 1.5% (v/v), in order to match that of the ligand solution. Ligands were dissolved in DMSO to a stock concentration of 10 mM, and then further diluted to 150–200 µM using buffer from the protein dialysis. The protein and ligand solutions were thoroughly degassed, and the protein solution was placed in a 1.4 mL cell of a VP ITC calorimeter (Malvern) with the experimental temperature set to 30 °C and differential power at 15 µCal s$^{-1}$. The respective ligands were injected at 300 s intervals after an initial delay of 180 s. Dilution enthalpy was calculated from post-saturation injections and subtracted from total heats, and the data were fit using a single-site model with the program OriginLab 7.5.

**Site-directed mutagenesis and TEVC recordings**. Missense mutations were introduced into rat cDNAs encoding GluN1-4a and GluN2B in the pGEM-HE vector using the QuikChange protocol (Stratagene) using the primers listed in Supplementary Table 2. These plasmids were then linearized with NotI and used as templates for cRNA synthesis (Ambion). A total mass of 5–10 ng cRNA at a 1:2 weight ratio of GluN1-4a and GluN2B transcripts was co-injected into defolliculated Xenopus laevis oocytes (EcoCyte Bioscience). The oocytes were incubated in Barth's solution containing 88 mM NaCl, 2.4 mM NaHCO$_3$, 1 mM KCl, 0.33 mM Ca(NO$_3$)$_2$, 0.41 mM CaCl$_2$, 0.82 mM MgSO$_4$, and 5 mM HEPES, at a final pH of 7.4, supplemented with 0.1 mg mL$^{-1}$ gentamicin sulfate, 1 µg mL$^{-1}$ streptomycin, and 1 U mL$^{-1}$ penicillin at 15–19 °C. TEVC current recordings were performed 2–4 days after injection using an extracellular recording solution containing 90 mM NaCl, 10 mM HEPES, 1 mM KCl, 0.5 mM BaCl$_2$, and 0.01 mM EDTA at either pH 6.9 or pH 7.6[39]. Voltage clamp was achieved and current responses were recorded at a holding potential of −40 mV with a two-electrode voltage clamp amplifier (Warner) at room temperature (~23 °C). Currents were low-pass filtered at 10 Hz and digitized at 20 Hz by LabWindows/CVI (National Instruments). Maximal effective concentrations of glutamate and glycine (100 and 30 µM, respectively) were used in all recordings. Potency (IC$_{50}$ values) for NMDA receptor inhibitors

were obtained by fitting the concentration–response curves using

$$\text{Response } (\%) = (100 - \text{minimum})/(1 - ([\text{modulator}]/\text{IC}_{50})^{nH}) + \text{minimum},$$
$$(1)$$

where IC$_{50}$ is the concentration of modulator that generates a half-maximal effect, $nH$ is the Hill slope, and minimum is the degree of residual inhibition at a saturating concentration of the NMDA receptor inhibitors.

**Reporting Summary**. Further information on experimental design is available in the Nature Research Reporting Summary linked to this article.

## Data availability

Data supporting the findings of this manuscript are available from the corresponding author upon reasonable request. Atomic coordinates and structure factors have been deposited in the Protein Databank under accession codes 6E7R (ATD heterodimer with 93-4), 6E7S (93-5), 6E7T (93-6), 6E7U (93-31), 6E7V (93-88), 6E7W (93-97), and 6E7X (93-115). Source files for Figs. 1b, 4c–j, and 2, and Supplementary Figures 1a, b and 7a–c are provided as a Source Data file.

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

## Acknowledgements

This work was supported by grants from the National Institutes of Health to M.C.R. (NS093753), H.Y. (HD082373), S.F.T. (NS036654 and NS065371), and H.F. (MH085926 and GM105730), Robertson funds at Cold Spring Harbor Laboratory, Austin's purpose, and Stanley Institute of cognitive genomics (all to H.F.). We would like to thank the staff at the National Synchrotron Light Source II, Beam Line 17-ID-1 at Brookhaven National Laboratory, and the Advanced Photon Source, Beam Line 23-ID-B at Argonne National Laboratory for assistance in X-ray crystallographic data collection. We would also like to thank Noriko Simorowski, Sukhan Kim, Jing Zhang, and Phuong Le for excellent technical assistance. Darryl Pappin is thanked for insightful discussions on structure-based mechanisms. We thank Janssen Research and Development, LLC, for providing pKa analysis for 93-31.

## Author contributions

M.C.R., D.C.L., D.S.M., S.J.M., H.Y., Z.Z., Y.A.T., S.F.T. and H.F. designed experiments. M.C.R. expressed, purified, and crystallized proteins, collected and processed X-ray diffraction data, and performed and processed ITC experiments. Y.A.T., D.S.M. and D.C.L. synthesized and purified compounds. H.Y., Z.Z., S.J.M. and S.F.T. performed TEVC experiments and analyzed electrophysiology data. All authors wrote the manuscript.
