## [Peer Review File · Nature Communications]

Reviewers' Comments:

Reviewer #1:

Remarks to the Author:

NMDA receptors are the major contributors to the excitatory neurotransmission that play an important role in learning and memory. They are implicated in neurotoxicity accompanying ischemic injuries and stroke and a number of neurological diseases and disorders, including epilepsy, depression and schizophrenia. Despite NMDA receptors represent a key target for drug development, success over the last three decades in making useful drugs has been limited. One of the key missing components for the efficient drug design is structural information about the mechanisms and binding sites of the known NMDA receptor modulators. The work of Regan et al. presents such molecular details for the new perspective class of NMDA receptor negative allosteric inhibitors that belong to the 93-series.

The manuscript illustrates the results of crystallographic, ITC, electrophysiology and mutagenesis experiments that together identify the molecular bases of 93-series interaction with the GluN1/GluN2B ATD. The study reveals elements of the 93-series structures that determine their binding affinity in a pH-dependent manner as well as the elements of ATD that are critical to both the 93-series binding and the type of the allosteric effect. Given the importance of developing NMDA receptor modulators specific to certain combinations of NMDA receptor subunits that can be brain region-specifically targeted in select neuropathologies, especially at characteristic to ischemia lower pH, this study undoubtedly presents an advance in our understanding of the molecular bases of NMDA receptor modulation that will be instrumental for the development of a future generation of drugs for treatment of ischemic injuries and a large number of devastating neurological diseases and disorders.

The described work includes an impressive amount of crystallographic, ITC, mutagenesis and electrophysiological data. The experimental results are of high quality and interpretations are just and fair. I only have a few minor comments and suggestions to further improve this sophisticated, important and well-written manuscript:

1. Page 3, line 4 of the first paragraph. 'mechanism' should be changed to 'mechanisms'.
2. Page 5, lines 4-5. Since errors and number of measurements are not presented here, please make a reference to Table 2 (the same for the Figure 1C legend).
3. Page 5, lines 6-7. To be correct, the affinities are for the ATD, not for the entire receptor. For example, the differences between the pH boost values for ITC and TEVC measurements might in part be due to differences in affinity.
4. Page 6, lines 6-9 of the first paragraph. Please make it more clear in the Methods section what exactly was done differently to achieve better diffraction.
5. Page 7, line 9. Since the binding site is deemed 'hydrophobic cage', it would be nice to illustrate its hydrophobic character by showing an example of the inhibitor (e.g., in stick representation) with the protein as a surface colored according to the electrostatic potential.
6. Page 8, line 6 (similarly, Page 9, lines 6 and 9). Please fix typo in 'van der Waals'.
7. It is not clear why the ITC and TEVC experiments are done with the GluN1a-GluN2B, while crystallographic experiments with GluN1b-GluN2B. It is understandable that GluN1a-GluN2B did not crystallize but why ITC and TEVC experiments were not done with GluN1b-GluN2B for consistency?
8. Page 12, 3rd line from the bottom. Please delete 'very'.
9. Page 13, 3rd line from the bottom. 'On' should probably be 'for'.
10. In Discussion, it would be great to have a brief discussion for the general reader on what the authors think is happening mechanistically when mutations in ATD inhibit or potentiate NMDA receptor channel-mediated currents.
11. Page 19, first line of the first paragraph. Please add 'purity' after '>95%'.
12. Page 19, second line of the first paragraph. 'compound' should be changed to 'compounds'.
13. Page 19, line 5 of the first paragraph. Please fix typo in 'hemitartrate'.
14. It would be beneficial to have a supplementary figure illustrating the efficiency of TEV and EndoF1 digests of GluN1a-GluN2B ATD on SDS-PAGE.
15. The potency was fitted with the equation on page 23, which includes the nH parameter. Since

the nH values are not presented, please add to the Table 2 footnotes a comment on the range of nH values.

Reviewer #2:

Remarks to the Author:

In the present study the authors show new data on compounds (93-series) whose activity against the GluN1-GluN2B receptor is highly pH sensitive, being more effective at antagonizing activity at lower pH. These compounds have potential in the clinic where during stroke affected areas can become acidic and the prominent role of GluN2B subunit in inducing cell death under such conditions. Although some of these compounds have been published on previously, as I understand, a number of new compounds are tested. However the strength of the manuscript are the novel mechanistic insights into how the pH sensitivity of these compounds arise which will have broad interest to a variety of fields.

The authors use a number of techniques including ITC, electrophysiology and structures to characterize and define the interaction of the 93-series compounds with the N-terminal domain of GluN1-GluN2B. Interestingly they identify a unique hydrophobic binding pocket (an extension of the ifenprodil pocket) for these compounds in the N-terminal domain and that the volume of a N-alkyl moiety of the 93-series compounds determines the pH sensitivity.

Overall the manuscript is presented extremely well. The figures are readable. The interpretation of the results is appropriate. The data are extremely intriguing and will lay the foundation for developing new compounds with even more specificity. Still I have a few suggestions that would make the conclusions of the manuscript stronger.

Comments

1. 'pH boost' As the Authors note the relative pH boost for the intact receptor using electrophysiology (around 7.5) is considerably greater than that observed for the isolated NTD domain using ITC (around 1.5). The authors make a number of arguments in the Discussion for why this might be. Although these arguments may be reasonable, some of these ideas could be tested especially on some of the mutations that have altered the novel binding pockets.

For example is the ratio of the ITC shift the same for 93-31 in the wild type background against the GluN2B(Y109W). One would expect that if the ITC and electrophysiology experiments are measuring the same feature that at least the relative ratio of changes should be consistent. Not all of the constructs need to be tested but to see that there is some rational level of consistency between ITC numbers and electrophysiological numbers, at least in terms of relative differences, would give more confidence that the two indices are measuring the same biophysical parameter. It would also be informative to do ITC experiments on the mutations that potentiate with pH changes.

2. For the ITC experiments, I was confused as to the GluN1 construct that was used.

3. Table 1. I believe some numbers are missing – exponentials for ifenprodil (e.g., -1.48 +/- 265??). Also the table is hard to extract quantitative information because of so many significant digits to the numbers. Is there really any reason to go beyond 2 digits past the decimal. Even this might not be needed. For example the pH boost could be shown as (for 93-97) as 2.1 +/- 0.1. Table 2 on the other hand is much easier to read. Anyway, it would be much easier to read the Table 1.

4. Is not there some way to measure more details on the electrophysiology. Although the IC50 is informative they do not reveal much about how the 93-series compounds are acting. Do they change glutamate affinity for example?

Reviewer #3:

Remarks to the Author:

The study by Regan and colleagues addressed the structural basis of pH-sensitive inhibition of GluN2B NMDARs. This work is based on the previous finding from Traynelis' group of identification of pH-sensitive GluN2B inhibitors. This topic is of importance for our understanding of the basic structural and function relationship of NMDARs (receptor to a larger extent) and likely has important translational implications for treating neurological diseases. The authors used multiple approaches to address this question. The basic findings are clear and interesting, clearly represents an important step towards a deeper understanding of NMDAR function and regulation. That being said, there are a few major questions/issues need to be clarified. In addition, I feel the presentation of the results can be improved and clarified by adding more information and explanation (at least for the key results), in order for a better and easier understanding of the results by the wide readership of Nature Communications.

Major:

1. Is there any difference between GluN1a and GluN1b that is critical to the results of this study? Since TEVC was performed on NMDARs containing the former while the structural analysis used the latter. It is important to repeat the key results using TEVC on GluN1b-GluN2B, simple saying that ifenprodil has similar binding affinity on these two is not sufficient.
2. Why doesn't 93-31 provide the largest pH-boost if its N-alkyl group matches most closely the hydrophobic cage? The authors should provide at least an explanation if not experimental data.
3. Why does GluN2B(D136A) have a pH-boost of 11, almost doubling the values of the other mutations?
4. It is stated on P 9, "the above mutagenesis studies indicate that all of the residues contributing to hydrophobic interactions with the N-alkyl group of the 93-31 are involved in controlling efficacy, potency, and pH-boost". Is Asp136 part of this hydrophobic cage (it caps the hydrophobic binding pocket) but mutation of it has virtually no effect?
5. For the effects seen in Fig. 4H and I, are there any compounds known to achieve this pH-dependent inhibition and potentiation? If so, do these compounds also have N-alkyl group similar to those in the 93-series?
6. Since the potentiation effect is claimed to be facilitated by the N-alkyl group (P. 10), is then the N-alkyl group selectively required for the pH-sensitivity of the 93-series compounds, or is it also involved in determining the nature of the modulation (inhibition vs. potentiation)? I am confused by the stated contribution of His134 to pH-sensitivity and controlling inhibition vs. potentiation.
7. The authors should discuss what controls the conversion from inhibition to potentiation, and how does it happen while preserving the pH-sensitivity? This is also related to the question of the contributions of N-alkyl group in 93-series.

Minor:

1. How much is the difference in the reduction in pH boost between TEVC and ITC experiments?
2. "... the binding and downstream action of" (P. 13).. The authors need to be more specific on "downstream"

3. P. 5, When discussing "...93-4, N-ethyl 93-5, N-propyl 93-6, and Nisopropyl 93-115 (Fig. 1a, 2c, d)..." on P.5, the only relevant figure is 2c, not fig. 2d.

4. Legend of Fig. 3."... f, g Overlay of 93-31 and ifenprodil25 (yellow, PDB code: 3QEL)". It should be f only.

.

Dear Editors & Editorial Staff,

We recently received comments on our submitted manuscript, “Structural elements of pH-sensitive inhibitor binding site in NMDA receptors” (Ref. No. NCOMMS-18-28073). We wish to thank the reviewers and Nature Communications staff for their thoughtful comments, which we have addressed below. For clarity, we directly respond to each comment in turn, with the reviewers’ comments in italic.

Reviewer #1-3

One common issue we saw in the reviewers’ comments is the confusion associated with running functional assays on the GluN1a splicing variant while running the structural studies on the GluN1b splicing variant. The 93-series compounds were originally found and characterized on the GluN1a splicing variant by Yuan et al (2015) using electrophysiology, therefore, we decided to conduct functional assays (direct binding assay by ITC and mutagenesis/electrophysiology) on the GluN1a splicing variant to permit comparison with the previous study. The reason we conducted high-resolution x-ray crystallography on the GluN1b splicing variant is because of the technical issue associated with acquiring well diffracting crystals with the GluN1a-GluN2B ATD the reason of which is not known to date. However, there is no difference in the overall architecture between GluN1a-GluN2B ATD and GluN1b-GluN2B ATD. The crystal structure of the intact GluN1a-GluN2B NMDA receptors (Karakas and Furukawa, 2014) showed that the overall structure of the GluN1a-GluN2B ATD and the binding mode of ifenprodil is almost identical to the GluN1b-

GluN2B ATD structure (Karakas et al., 2011; Regan et al., 2018). The 21 amino acid residues encoded by exon 5 is also located further away from the binding site of the allosteric modulators (Karakas et al., 2011) including 93-31 (please see the Figure. Arrow is where the exon5-encoded motif would be located and sticks are 93-31 and the binding residues. Note that electron density for the exon5-encoded motif was not resolved here), thus, the presence of exon5 does not alter the binding mode of the compounds. Therefore, we believe that the compound

binding information gained from the crystal structures of the GluN1b-GluN2B ATD in the present study is directly applicable to study the GluN1a splicing variant. The above justification is incorporated in the revised manuscript. We have also added new data (*see* Supplemental Figure 1b) confirming that 93-31 has the same IC₅₀ values and pH sensitivity for receptors that contain exon5 as those that lack exon5, consistent with the idea that exon5 is distant to the 93-31 binding site and not involved in its pH-sensitivity.

Reviewer #1 (Remarks to the Author):

1. Page 3, line 4 of the first paragraph. ‘mechanism’ should be changed to ‘mechanisms’.

This has been addressed in the text.

2. Page 5, lines 4-5. Since errors and number of measurements are not presented here, please make a reference to Table 2 (the same for the Figure 1C legend).

The appropriate errors have been incorporated into the text, and there is now a reference to Table 2 both in the text and the Figure 1C legend.

3. Page 5, lines 6-7. To be correct, the affinities are for the ATD, not for the entire receptor. For example, the differences between the pH boost values for ITC and TEVC measurements might in part be due to differences in affinity.

This has been clarified in the text.

4. Page 6, lines 6-9 of the first paragraph. Please make it more clear in the Methods section what exactly was done differently to achieve better diffraction.

We have expanded upon this in the Methods section.

5. Page 7, line 9. Since the binding site is deemed 'hydrophobic cage', it would be nice to illustrate its hydrophobic character by showing an example of the inhibitor (e.g., in stick representation) with the protein as a surface colored according to the electrostatic potential.

We added the suggested structural presentation in the revised Fig. 4a.

6. Page 8, line 6 (similarly, Page 9, lines 6 and 9). Please fix typo in 'van der Waals'.

This has been corrected in the text.

7. It is not clear why the ITC and TEVC experiments are done with the GluN1a-GluN2B, while crystallographic experiments with GluN1b-GluN2B. It is understandable that GluN1a-GluN2B did not crystallize but why ITC and TEVC experiments were not done with GluN1b-GluN2B for consistency?

We initially sought to perform all experiments on the GluN1a splice variant, thus, we conducted biophysical characterization including all ITC and TEVC experiments on this construct. However, despite considerable efforts, we were unable to crystallize the GluN1a-GluN2B ATD heterodimer to high resolution, thus, we visualized the ligand-binding site by x-ray crystallography on the GluN1b-GluN2B ATD. The GluN1a-GluN2B ATD structure from our previous intact NMDA receptor structure (PDB: 4PE5) and the GluN1b-GluN2B ATD are virtually identical and the exon5-encoded motif in GluN1b does not have a physical impact on binding of allosteric inhibitors. Furthermore, our new data showing virtually identical pH sensitivity of 93-31 in the absence or presence of exon5 strengthens our conclusions.

8. Page 12, 3rd line from the bottom. Please delete 'very'.

This has been addressed in the text.

9. Page 13, 3rd line from the bottom. 'On' should probably be 'for'.

In this case we are referring to the ligand itself, and the site to be modified is on the ligand.

10. In Discussion, it would be great to have a brief discussion for the general reader on what the authors think is happening mechanistically when mutations in ATD inhibit or potentiate NMDA receptor channel-mediated currents.

Our current model of activation involves opening of the GluN2B ATD bi-lobe and rearrangement of the GluN1-GluN2B ATD dimer interface, which are translated to rearrangement of the GluN1-GluN2B LBD dimers in a way to create tension in the channel gate (Tajima et al., 2016). Since the "hydrophobic cage" is located in proximity to the GluN1-GluN2B dimer interface, we tentatively believe that the GluN1 Tyr109Trp or His134Ala mutants, when the 93-31 is bound, have a tendency to take the 'active' conformation. Of course, the above is just a speculation at this point and testing this hypothesis requires substantial crystallographic efforts on the mutant receptors, which will take more time. At this point, we would like to modestly suggest a mechanism involving the GluN1-GluN2B ATD dimer interface by citing Tajima et al 2016. Text is added to the "Discussion" section to address the above points.

11. Page 19, first line of the first paragraph. Please add 'purity' after '>95%'.

This has been corrected in the text.

12. Page 19, second line of the first paragraph. 'compound' should be changed to 'compounds'.

This has been corrected in the text.

13. Page 19, line 5 of the first paragraph. Please fix typo in 'hemitartrate'.

This has been corrected in the text.

14. It would be beneficial to have a supplementary figure illustrating the efficiency of TEV and EndoF1 digests of GluN1a-GluN2B ATD on SDS-PAGE.

Below, we are providing an SDS-PAGE gel of the GluN1/GluN2B ATD purification process for the reviewer's examination:

- Lane 1: Endo F1
 Lane 2: BioRad Protein Standards
 Lane 3: Conditioned media, GluN1a/GluN2B ATD fusion construct expression.
 Lane 4: Conditioned media, GluN1b/GluN2B ATD fusion construct expression.
 Lane 5: Cobalt-charge NTA resin GluN1a/GluN2B ATD fusion elution.
 Lane 6: Cobalt-charge NTA resin GluN1b/GluN2B ATD fusion elution.
 Lane 7: Overnight TEV and Endo F1 digest of GluN1a/GluN2B ATD fusion construct.
 Lane 8: Overnight TEV and Endo F1 digest of GluN1b/GluN2B ATD fusion construct.
 Lane 9: Flow-through of CoNTA column after GluN1a/GluN2B digest to remove TEV, Endo F1, and GluN1a/GluN2B ATD linker.
 Lane 10: Flow-through of CoNTA column after GluN1b/GluN2B digest to remove TEV & Endo F1.
 Lane 11: Elution of TEV & Endo F1 trapped by CoNTA column (GluN1a/GluN2B ATD sample)
 Lane 12: Elution of TEV & Endo F1 trapped by CoNTA column (GluN1b/GluN2B ATD sample)

15. The potency was fitted with the equation on page 23, which includes the nH parameter. Since the nH values are not presented, please add to the Table 2 footnotes a comment on the range of nH values.

Values range from 0.4 – 1.3. As suggested, we have added the nH values to Table 2.

Reviewer #2 (Remarks to the Author):

Comments

1. 'pH boost' As the Authors note the relative pH boost for the intact receptor using electrophysiology (around 7.5) is considerably greater than that observed for the isolated NTD domain using ITC (around 1.5). The authors make a number of arguments in the Discussion for

why this might be. Although these arguments may be reasonable, some of these ideas could be tested especially on some of the mutations that have altered the novel binding pockets. For example is the ratio of the ITC shift the same for 93-31 in the wild type background against the GluN2B(Y109W). One would expect that if the ITC and electrophysiology experiments are measuring the same feature that at least the relative ratio of changes should be consistent. Not all of the constructs need to be tested but to see that there is some rational level of consistency between ITC numbers and electrophysiological numbers, at least in terms of relative differences, would give more confidence that the two indices are measuring the same biophysical parameter. It would also be informative to do ITC experiments on the mutations that potentiation with pH changes.

We agree that this would be an interesting avenue to explore, and we thank the reviewer for this insight. Our mutagenesis studies showed dramatic shifts in inhibition potency (measured by TEVC) when binding residues are mutated. What this means is that we can no longer precisely measure potency by ITC that is not ideal for measuring affinity beyond 1 μ M Kd. The pH-boost of the 93-series compounds is an extremely sensitive parameter, thus, precise determination of the binding parameter (Kd) is crucial. Thus, we are technically limited to precisely measure Kd values of the mutants in this case. In the similar view to the reviewer, we instead tested different 93-series compounds using ITC and correlated the Kd values with the measured IC50 values in electrophysiology. As the reviewer knows, the throughput of the ITC experiment is low due to large protein amount necessary for running such experiments. Adding to this difficulty is the fact that the protein expression yield is low for the GluN1-GluN2B ATD dimer proteins (e.g. we need to co-express in mammalian or insect cells and isolate the heterodimeric species). Thus, running triplicate experiments on each 18 conditions in Table 1 was already a difficult and time-consuming task. With respect to running an ITC experiment on ‘potentiating’ mutant, GluN2B (Y109W), we, too, think it’s interesting. We are also aware that it would not distinguish potentiation from inhibition since ITC only measures energies or thermodynamic parameters (ΔG , ΔH , and ΔS). Indeed, we are testing the expression of this mutant in the context of ATD, however, our preliminary data indicates that the expression level is even lower than the WT. Regenerating virus, optimizing purification protocol, and eventually performing the ITC experiments would take months of efforts and is unfortunately outside of our time frame and scope. We will continue to work hard toward answering the reviewer’s question in a separate set of work beyond this one.

2. For the ITC experiments, I was confused as to the GluN1 construct that was used.

We performed ITC experiments using primarily the GluN1a/GluN2B ATD construct, i.e., all titrations of 93-series ligands were performed on GluN1a/GluN2B. However, to eliminate the possibility of an effect resulting from the presence of the alternatively-splice Exon 5 residues (GluN1b), we also performed ITC with GluN1b and Ifenprodil. All values listed in Table 1 are for GluN1a/GluN2B, except where noted for the Ifenprodil experiment. We have modified the table to make this clear

3. Table 1. I believe some number are missing – exponentials for ifenprodil (e.g., -1.48 +/- 265??). Also the table is hard to extract quantitative information because of so many significant digits to the numbers. Is there really any reason to go beyond 2 digits past the decimal. Even this might not be needed. For example the pH boost could be shown as (for 93-97) as 2.1 +/- 0.1. Table 2 on the other hand is much easier to read. Anyway, it would be much easier to read the Table 1.

We thank the reviewer for the thorough assessment of Table 1. This has been corrected in the new Table 1.

4. Is not there some way to measure more details on the electrophysiology. Although the IC50 is informative they do not reveal much about how the 93-series compounds are acting. Do they change glutamate affinity for example?

The question of mechanism will ultimately be answered by structural analysis on the full-length NMDA receptors in both inhibited and non-inhibited conformations (at two pH values), which is beyond the scope of the current study. Having said that, we have addressed the mechanism of allosteric inhibition by ifenprodil, the prototypical allosteric modulator compound binding to the same site as the 93-series, previously (Tajima et al., 2016). Our current model of ATD-mediated allosteric inhibition involves closure of the GluN2B ATD bi-lobe to prevent rearrangement of the GluN1-GluN2B ATD dimer interface, which also prevents the rearrangement of the GluN1-GluN2B LBD dimers coupled to the channel gating. Since the overall structure and the conformation of the GluN1-GluN2B ATD complexed to the 93-31 series is almost identical to the one bound to ifenprodil, we predict that the mechanism of the 93-31 inhibition involves highly similar mechanism. Regarding glutamate potency in the presence of 93-series ligands, we have added new data showing no change in glutamate EC₅₀ in the presence of a half maximally inhibiting concentration of 93-31 at two pH values.

Reviewer #3 (Remarks to the Author):

Major:

1. Is there any difference between GluN1a and GluN1b that is critical to the results of this study? Since TEVC was performed on NMDARs containing the former while the structural analysis used the latter. It is important to repeat the key results using TEVC on GluN1b-GluN2B, simple saying that ifenprodil has similar binding affinity on these two is not sufficient.

We previously crystallized the intact, inhibited GluN1a/GluN2B NMDA receptor (PDB ID: 4PE5) in the presence of ifenprodil. Here, we found the orientation of the GluN1a/GluN2B ATD heterodimer to be nearly identical to the GluN1b/GluN2B ATD heterodimer that we report here. We also have added new data showing 93-31 inhibits GluN1-1b/GluN2B receptors with identical potency and pH sensitivity as GluN1-1a/GluN2B, confirming that exon5 does not influence the mechanism of pH sensitivity of the 93-series compounds.

2. Why doesn't 93-31 provide the largest pH-boost if its N-alkyl group matches most closely the hydrophobic cage? The authors should provide at least an explanation if not experimental data.

At the level of electrophysiology, 93-31 has the highest pH-boost as shown by Yuan et al (2015). The pH-boost of the direct binding as assessed by ITC shows equivalent or more pH-boost for 93-97 or 93-88 (another isomer of 93-31). However, these two compounds (93-97, 93-88) were not as potent as 93-31, probably due to protrusion of the hydrophobic N-alkyl group from the hydrophobic cage that may affect conformational dynamics of the ATDs or free energy of the

bound ligand. We added some text explaining this possibility. To obtain data that speaks to this question, one may need to complete structural studies on the intact NMDA receptors complexed to different 93-series compounds that showed similar potencies but different geometries of the alkyl substitution. At this point, perturbations in alkyl group reduce potency, precluding the analysis sought. The effort to obtain structural data for full-length NMDA receptors bound to different 93-series ligands is beyond the scope of the current work.

3. Why does GluN2B(D136A) have a pH-boost of 11, almost doubling the values of the other mutations?

Our current hypothesis is that this mutation probably enhances the strength of hydrophobic interaction with the N-alkyl group. This point is incorporated in the text.

4. It is stated on P 9, “the above mutagenesis studies indicate that all of the residues contributing to hydrophobic interactions with the N-alkyl group of the 93-31 are involved in controlling efficacy, potency, and pH-boost”. Is Asp136 part of this hydrophobic cage (it caps the hydrophobic binding pocket) but mutation of it has virtually no effect?

Although Asp136 is not hydrophobic, we included it in the figure to give readers a sense of boundary between hydrophobic and hydrophilic regions. Asp136 is solvent accessible and minimally affects potency of inhibition and pH-boost. We omitted the description ‘cap’ to avoid confusion.

5. For the effects seen in Fig. 4H and I, are there any compounds known to achieve this pH-dependent inhibition and potentiation? If so, do these compounds also have N-alkyl group similar to those in the 93-series?

The 93-series is the first set of compounds we are aware of that shows strong pH dependence, as well as potentiation for these GluN1 mutations. Please note that the 93-31 acts as a potentiator only on the H134A and Y109W mutant. They always work as inhibitors on WT receptors.

6. Since the potentiation effect is claimed to be facilitated by the N-alkyl group (P. 10), is then the N-alkyl group selectively required for the pH-sensitivity of the 93-series compounds, or is it also involved in determining the nature of the modulation (inhibition vs. potentiation)? I am confused by the stated contribution of His134 to pH-sensitivity and controlling inhibition vs. potentiation.

We view the contribution of the ionization of His134 to the pH sensitivity as a separate phenomenon from the ability of a mutation at this position to produce potentiation.

The 93-series as well as the prototypical negative allosteric modulator, ifenprodil, always inhibit the WT GluN1-GluN2B NMDA receptors. The length (number of carbons) and nature (straight vs branched) of the N-alkyl group in the 93-series dictates the potency of inhibition and pH-sensitivity of inhibition on the WT GluN1-GluN2B NMDA receptors. We believe this is because the association/dissociation rates are pH dependent. GluN1 His134 is close to the hydrophobic cage, and is accessible to solvent, and thus will change its protonation states in with changing pH. The pKa of His134, which is ~6.0 in free solution, is ideally poised to maximally change the ionization

of His with a reduction in pH from 7.6 to 6.9. Whether His134 is charged or not determines how strongly it interacts with GluN1-Ile133, which is interacting with the N-alkyl group of the 93-31 series. Thus, the changing ionization of His134 with changing pH will contribute to the proton sensitivity of 93-31 by changing the nature of the hydrophobic cage.

In the course of this study, we found that the 93-series ‘potentiates’ two mutant GluN1-GluN2B NMDA receptors (GluN1-His134Ala or GluN1-Tyr109Trp), one of which had previously been described. This potentiation reflects some structural change in the receptor, not simply removal of the ionizable His residue. We expect that 93-31 adopts a unique pose when in complex with the two mutant receptors that leads to potentiation of receptor function by a yet-to-be determined mechanism. Ifenprodil lacks the N-alkyl group, does not engage the hydrophobic cage, and only minimally potentiates both mutant receptors, if at all.

We reorganized the text to more clearly deliver this point, as well as better separate the role of the ionizations of His134 in pH sensitivity and the role of His134Ala in converting inhibitors to potentiators.

7. The authors should discuss what controls the conversion from inhibition to potentiation, and how does it happen while preserving the pH-sensitivity? This is also related to the question of the contributions of N-alkyl group in 93-series.

Our current model of activation involves opening of the GluN2B ATD bi-lobe and rearrangement of the GluN1-GluN2B ATD dimer interface, which are translated to rearrangement of the GluN1-GluN2B LBD dimers in a way to create tension in the channel gate (Tajima et al., 2016). Since the “hydrophobic cage” is located in proximity to the GluN1-GluN2B dimer interface, we tentatively believe that the GluN1 Tyr109Trp or His134Ala mutants, when the 93-31 is bound, has a tendency to adopt the ‘active’ conformation to mediate potentiation rather than inhibition. That is, binding of 93-31 may facilitate opening of the GluN2B ATD bi-lobe and rearrange GluN1-GluN2B heterodimeric interface. Of course, the above is speculative at this point, and testing these hypotheses requires substantial structural biological efforts on the mutant full-length receptors, which will take more time. At this point, we cautiously suggest a mechanism involving the GluN1-GluN2B ATD dimer interface by citing Tajima et al 2016. Text is added to the “Discussion” section to address the above points.

Minor:

1. How much is the difference in the reduction in pH boost between TEVC and ITC experiments?

We are not sure what the reviewer means by “reduction.” All of the parameters including pH-boost are listed in Table 1 and 2.

2. “... the binding and downstream action of ...” (P. 13).. The authors need to be more specific on “downstream”

We agree with the reviewer. The text now read as follow: ‘... suggesting a portion of the effect is resulting from downstream aspects of how 93-31 produces channel inhibition perhaps involving conformational alteration at the LBD region.’

3. P. 5, When discussing “....93-4, N-ethyl 93-5, N-propyl 93-6, and Nisopropyl 93-115 (Fig. 1a, 2c, d)...” on P.5, the only relevant figure is 2c, not fig. 2d.

Fig. 2d plots pH-boost for 93-4, -5, -6, and -115, so we’d like to keep the text as it is.

4. Legend of Fig. 3.”... f, g Overlay of 93-31 and ifenprodil25 (yellow, PDB code: 3QEL)”. It should be f only.

We agree. Thanks for noticing.

References

- Karakas, E., and Furukawa, H. (2014). Crystal structure of a heterotetrameric NMDA receptor ion channel. *Science* *344*, 992-997.
- Karakas, E., Simorowski, N., and Furukawa, H. (2011). Subunit arrangement and phenylethanolamine binding in GluN1/GluN2B NMDA receptors. *Nature* *475*, 249-253.
- Regan, M.C., Grant, T., McDaniel, M.J., Karakas, E., Zhang, J., Traynelis, S.F., Grigorieff, N., and Furukawa, H. (2018). Structural Mechanism of Functional Modulation by Gene Splicing in NMDA Receptors. *Neuron* *98*, 521-529 e523.
- Tajima, N., Karakas, E., Grant, T., Simorowski, N., Diaz-Avalos, R., Grigorieff, N., and Furukawa, H. (2016). Activation of NMDA receptors and the mechanism of inhibition by ifenprodil. *Nature*.

Reviewers' Comments:

Reviewer #1:

Remarks to the Author:

The reviewers' comments were satisfactorily addressed and I have no further comments.

Reviewer #2:

Remarks to the Author:

The authors have largely addressed my previous concerns. I have no further comments.

Reviewer #3:

Remarks to the Author:

The authors have addressed all my questions and have done a wonderful job in revising the manuscript. I have no further questions.